# The Relationship between Jealousy and Mate Retention Strategies in Romantic Relationships among Women during the COVID-19 Pandemic

Paulina Degiuli, Lea Andreis * and Dario Vučenović 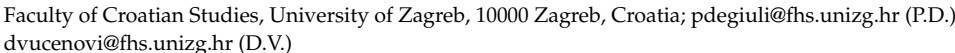

Faculty of Croatian Studies, University of Zagreb, 10000 Zagreb, Croatia; pdegiuli@fhs.unizg.hr (P.D.); dvucenovi@fhs.unizg.hr (D.V.)
* Correspondence: landreis@fhs.unizg.hr

**Abstract:** Jealousy and mate retention have received attention in research over the last few decades. Despite this, most of the research has examined male jealousy and male mate retention, emphasizing cost-inflicting behavior due to its role in relationships and domestic violence. The aim of this study was to investigate the relationship between jealousy and all mate retention strategies in romantic relationships among women during the COVID-19 pandemic. The sample consisted of 772 Croatian women aged 19 to 40 who were in a heterosexual relationship at the time. This study was conducted online, and the participants completed the Multidimensional Jealousy Scale and Mate Retention Inventory. The results showed that cognitive, emotional, and behavioral jealousy were positively correlated with all mate retention strategies, which indicates that a stronger experience of jealousy can be expected to result in more frequent use of all partner retention strategies. We also found that all three dimensions of jealousy and relationship length positively predicted both cost-inflicting and benefit-provisioning mate retention behavior, whereas age was a negative predictor of benefit-provisioning behavior only. The findings of this study suggest that, although jealousy can substantially explain interpersonally risky and damaging behavior in relationships, it can also explain affectionate and attentive behavior, to some extent.

**Keywords:** jealousy; partner retention strategies; romantic relationship; age; relationship length

## 1. Introduction

Individuals in romantic relationships encounter numerous challenges that can affect their relationships in various ways. One of those challenges can be the occurrence of romantic jealousy, which can be seen as an evolutionary adaptation. Although jealousy is often described as a negative emotion, when it occurs in small amounts, jealousy can contribute to positive outcomes in a romantic relationship, including expressing love and care toward a partner [1,2] and signaling a commitment to one's partner.

Given that jealousy is a complex construct that encompasses a range of reactions in response to the perception of a potential threat to a romantic relationship, Pfeiffer and Wong [3] divided these reactions into three components: the cognitive, emotional, and behavioral components of jealousy. The cognitive aspect of jealousy arises when a person becomes aware of a threat to their romantic relationship. Therefore, subjective experience of jealousy is strongly linked to the associated cognitive processes, as the person primarily must perceive their partner's affection toward another person to trigger jealousy. Emotional jealousy is defined [3,4] as the level of emotional distress a person feels in situations that threaten their relationship with their partner. Jealousy research has shown that two components of the affective state underlie emotional jealousy—the intensity and frequency of jealous responses [3,4]—which are inherent to emotions in general [5]. Behavioral jealousy is expressed through controlling behavior towards one's partner. Specifically, the behavioral expressions of jealousy are attempts to influence oneself,

one's partner, or the situation to preserve the romantic relationship, reduce insecurity, or maintain self-esteem [4]. Buunk [6] proposed a similar model that differentiates three types of jealousy: preventive, anxious, and reactive jealousy. Preventive jealousy relates to efforts to prevent a partner's interactions where infidelity may occur. Anxious jealousy refers to feeling upset or worried about the possibility of a partner's infidelity. Reactive jealousy is defined as a negative response to a partner's involvement, either emotional or sexual, with someone else. As Davis et al. argued [7], these two conceptualizations regarding the multidimensional nature of jealousy are congruent. More specifically, behavioral jealousy corresponds to preventive jealousy, cognitive jealousy to the anxious type of jealousy, and emotional jealousy to reactive jealousy.

Numerous studies have shown that women generally express more romantic jealousy than men in response to cues to emotional infidelity, whereas men express more romantic jealousy than women in response to cues to sexual infidelity [5,7,8] which is known as sexual dimorphism in response to jealousy-evoking situations [9]. These findings support the evolutionary hypothesis that, unlike men, women are certain of their biological connection to their offspring and are more concerned with getting help and resources from their partner, and emotional infidelity can signal that their partner is investing their resources elsewhere. Dissatisfaction with the relationship, lack of shared time, emotional neglect, a lack of partner attention, passivity in the relationship, and aggressive and rejecting communication can all provoke emotional jealousy [10].

Harris [11,12] argued that evolutionary theory and sexual dimorphism have several limitations when explaining jealousy and proposed a social–cognitive theory of jealousy based on the principles of social learning and social cognition. According to this view, jealousy occurs when an individual evaluates that a rival threatens any important aspect of a romantic relationship or poses a threat to the representation of the self [11–14]. The evaluation process of the threat to the relationship includes both how threatening the rival is and the meaning of the interaction between the partner and the rival. As jealousy is evoked when the rival is perceived as a threat and the interaction is regarded as flirtatious in one's culture, this is a general mechanism for detecting cues that evoke a jealousy response that considers the cultural and cognitive aspects of the situation [13]. Furthermore, Harris and Dolby [14] argued that the appraisal and attribution of a partner's behavior elicit a jealousy response, not the partner's behavior itself. Harris and Darby [14] also suggest that the emotion of jealousy evolves through multiple stages in accordance with Lazarus's cognitive theory of emotion, involving the cognitive appraisal of the situation, emotional reaction, and behavioral response. The initial stage involves the assessment of the situation and the rival, leading to a decision about the extent of the threat they pose to the romantic relationship, followed by a corresponding emotional reaction. The assumption is that an individual will experience jealousy if they perceive the rival as superior in areas that are crucial to the individual's self-esteem [14]. Another critique of sexual dimorphism is the interpretation of the results of studies that reveal gender differences in jealousy considering the specific type of infidelity. According to the social cognitive theory, men will automatically react more strongly to a partner's sexual infidelity because there is a widely held stereotypical belief that a woman engaging in a sexual relationship with another person also feels emotional closeness to that individual [13]. On the other hand, it is commonly believed that a man can have a sexual relationship without emotional attachment to another person. For this reason, women can express lower levels of sexual jealousy and higher levels of emotional jealousy because, according to this logic, a man's sexual infidelity may not be connected to emotional infidelity, but emotional infidelity in a man implies the existence of sexual infidelity [14]. In other words, men and women do not differ in the type of jealousy they feel, but in their personal interpretation of what a particular type of infidelity implies. De Visser et al. [9] also pointed out the limitations of the evolutionary perspective in a non-heterosexual context, since it cannot explain non-procreative sexual behavior, including same-sex sexual activity. Meta-analyses have supported the sexual dimorphism model demonstrating moderate effect sizes [11,15], and it should be noted that

Sagarin et al. [15] analyzed the effects of response style (forced choice or continuous), and hypothetical and real infidelity scenarios. They concluded that the observed sex difference in responses to emotional and sexual infidelity is not a measurement artifact, as was criticized [11]. On the other hand, Carpenter [14] conducted a meta-analysis that showed that both sexes were more upset by emotional than sexual jealousy, which is in line with social–cognitive theory. De Visser et al. [9] conducted a study on a population-representative sample that showed that, among heterosexual participants, the effects of sex differences predicted by sexual dimorphism were relatively modest, while social cognitive variables (e.g., education, religiosity, or previous infidelity) explained jealousy response variance that sex differences did not. Among non-heterosexual respondents, the effects of sex differences predicted by sexual dimorphism were the opposite, as lesbian and bisexual women were more likely than gay and bisexual men to respond that sexual infidelity was more distressing than emotional infidelity [9]. Carpenter [14] reported similar results regarding non-heterosexual individuals, concluding that "effects found were not rooted in the nature of participants' sexual orientation, but rather in their perceptions of the likely pattern of behavior appropriate to stereotyping of their partner's gender". Interestingly, when considering non-monogamous relationships, both heterosexual and non-heterosexual, if it is consensual (e.g., swinging partners) or partners have more permissive attitudes, sexual infidelity is considered less threatening than emotional [9,16]. Thus, attitudes or agreements between partners can shape jealousy responses, which is in line with social-cognitive theory. Regarding cross-cultural studies, gender differences in jealousy responses to emotional and sexual infidelity support the evolutionary perspective; nevertheless, culture produces variations in the extent to which individuals feel upset when encountering emotional and sexual infidelity [17,18]. Regarding jealousy expression, individualism/collectivism, masculinity/femininity, and egocentric thinking are cultural variables that can explain the observed differences between cultures [19]. Specifically, jealousy is more expressed in individualistic, masculine, and egocentric thinking cultures [19] because it is considered a socially (and culturally) accepted behavior. Zandbergen and Brown [18] found culture to be a more important predictor than gender when considering the jealous response to sexual infidelity, while gender was more important than culture for predicting the jealous response to emotional infidelity. It is also worth mentioning that cultural and social norms shape one's attitudes toward love and sex, which are entwined with the socialization of gender roles [11]. Social norms regarding sexual activity and personal attitudes about the importance of sex are more restricted for women than men. Premarital sex and extramarital sexual activities are regarded as more socially inappropriate for women, even in Western modern societies [11,18], while sexual activity is more important for male self-esteem [11]. This could explain why women are more susceptible to cues of emotional infidelity, while men have a greater jealousy response to sexual infidelity. This suggests that both gender and cultural norms can affect the way jealousy is expressed in romantic relationships. Hence, it can be concluded, as de Visser et al. [9]. Pointed out, that evolutionary and social–cognitive perspectives are complementary, rather than contradictory when explaining jealousy's underlying mechanisms.

Individuals' past experiences of attachment have a complex relationship with jealousy. Securely attached individuals tend to have a positive self-image, longer lasting and more successful romantic relationships, and lower levels of distrust [14,20]. Felmlee et al. [21] argued that the level of jealousy, especially in girls, increases during adolescence. Jealousy manifests not only in romantic relationships, but also in friendships, which are crucial for the process of self-discovery. There is a fear of the entry of a "third party" into the friendship, potentially disrupting the existing harmony. Research by Voulgaridou and Kokkinos [22] indicated a negative correlation between the secure attachment of adolescent girls to their parents and the level of jealousy in their friendships.

It is generally believed that insecurely attached individuals are more prone to experiencing negative emotions, including jealousy, as a consequence of inconsistent and rejecting responses from their primary caregivers. This assertion is supported by the findings of Mur-

phy et al. [23], where the results showed that adolescents securely attached to their parents exhibited lower levels of jealousy than those with anxious–avoidant and especially those with anxious–resistant attachment. This trend persists into young adulthood, as confirmed by the findings of Ross and Fuertes [24], indicating a connection between attachment to the father and better social skills, as well as easier emotional adjustment.

There is a very small number of studies examining the direct relationship between attachment to the father and female jealousy; however, valuable findings are provided by a study conducted by van Brummen-Girigori et al. [25]. The aim of this research was to investigate whether women whose fathers left during their upbringing experience higher levels of romantic jealousy compared with women who grew up in the presence of their fathers. The results indicated significantly higher levels of anxious and preventive jealousy in women whose fathers had left. These findings can be explained by the inability to form a secure attachment with the absent caregiver (in this case, the father), which is negatively associated with experiencing and expressing jealousy in both friendships and romantic relationships. Hence, father abandonment in childhood has a significant impact on a woman's relationship with men later in life, making women hyper-vigilant, resulting in strong reactions to any clue of possible partner infidelity.

Although attraction and the selection of a romantic partner are crucial for human reproduction, maintaining an existing intimate relationship is often necessary to fulfill promises of reproductive effort [26]. De Miguel and Buss [27] consider retaining a partner to be an important issue in romantic relationships, precisely because of the attempt to take a partner away from a third party, infidelity, and the risk of leaving an intimate relationship. Buss [26] defined such behavior as mate retention strategies and identified 104 such actions, which are divided into 19 tactics. These 19 partner retention tactics were grouped into 5 specific categories, which are direct guarding (e.g., checking on one's partner), intersexual negative inducements (e.g., flirting with someone else in the presence of one's partner), positive inducements (e.g., spending a large amount of money on one's partner), intrasexual negative inducements (e.g., derogation of a potential rival), and public signals of possession (e.g., kissing one's partner in the presence of others) [26]. In addition to the specific mate retention strategies, the categories can also be grouped into cost-inflicting strategies, which include direct guarding, intersexual negative inducements, intrasexual negative inducements, and benefit-provisioning strategies, which include positive inducements and public signals of possession. Cost-inflicting mate retention strategies are low-cost and high-risk strategies because they involve investing fewer resources, but taking a higher risk of losing a partner if not employed, which implies direct and indirect aggressive, controlling, and interpersonally damaging behavior. They are also risky due to the increased possibility of retaliation from either a partner or a rival, relationship termination, and a decrease in relationship satisfaction [28]. Benefit-provisioning strategies are high-cost and low-risk strategies, characterized by investing more resources, but their effect on retaining a partner is relatively low [29]. They also imply affectionate and attentive behavior toward the partner and are linked to positive relationship outcomes, such as increased satisfaction with the relationship and perceived higher relationship quality, as well as decreased odds of separation [28].

Sexual jealousy and partner retention are considered the most obvious partner retention tactics, but many tactics can be much more subtle. Some of them include distracting one's partner from a potential rival, portraying the rival as less attractive or desirable, buying gifts for one's partner, etc. It is also important to consider a person's age, as it is known that men are reproductively capable from puberty to old age, while women's fertility peaks in their mid-20s and significantly declines over time, especially in their late-40s [8,30]. Therefore, younger women are more desirable partners due to their reproductive potential and are potentially more desirable for "stealing". Thus, men married to younger, more attractive women should devote more time to partner retention strategies, as should women married to men with many resources [30]. Consequently, women's partner retention strategies may include, for example, improving their appearance [27]. Holden et al.'s findings [29]

showed that men, unlike women, use mate retention more frequently when having a younger and more attractive partner. Similarly, women's but not men's, mate retention was positively associated with the partner's income and status. De Miguel and Buss [27] found that younger women generally used all partner retention strategies more frequently than older women. However, they also showed that women in more committed relationships (engaged or married) usually used commitment manipulation (e.g., discussing pregnancy), displaying resources, improving appearance, love, and care, and verbal signs of possession more often than women in less committed relationships. Davies et al. [7] found a positive correlation between relationship length and both cost-inflicting and benefit-provisioning mate retention strategies, which implies that women in more committed relationships are also more motivated to use both mate retention strategies.

It can be assumed that a higher level of jealousy is positively associated with the frequency of using partner retention strategies [8,31]. Simply provoking jealousy can help to identify individuals or situations that may pose a threat to an existing romantic relationship. Identifying a threatening individual or situation can encourage the use of partner retention strategies designed to strengthen the romantic relationship or repress rivals [32]. Additionally, jealousy can have positive effects on the relationship, such as admiration for the partner and showing strong feelings for them. However, most previous studies have linked jealousy to various cost-inflicting partner retention strategies, including aggression toward the partner or rival and inflicting negative outcomes on the romantic relationship [33]. This is in line with the dynamic functional model of jealousy proposed by Chung and Harris [34], in which jealousy is a motivational state to secure relationships and block rivals. Jealousy emerges if an imaginary or actual rival is evaluated as a threat to either the relationship or the self. Once jealousy is elicited, it can manifest in behavior, emotion, cognition, physiological responses, or continuous appraisals, all of which can have beneficial or detrimental consequences to the maintenance of the relationship. Once again, as is the case with the underlying mechanism of jealousy, evolutionary and social–cognitive perspectives are complementary when considering outcomes once jealousy is elicited. In Conar's study [10], a positive correlation was found between all partner retention strategies (direct guarding, intersexual negative inducements, positive inducements, public signs of possession, and intrasexual negative inducements) and all dimensions of jealousy (cognitive, emotional, and behavioral). In the same study, the cognitive and behavioral dimensions of jealousy were positive predictors of both cost-inflicting and benefit-provisioning partner retention strategies, while relationship length was a negative one for benefit-provisioning strategies. As Conar's study [10] did not consider sex differences, it is worth mentioning that Davis et al. [7] found that women who were more prone to reactive (i.e., emotional) jealousy were also more motivated to use both cost-inflicting and benefit-provisioning mate retention strategies, whereas anxious (i.e., cognitive) and preventive (i.e., behavioral) jealousy positively predicted only cost-inflicting mate retention. Findings that emotional jealousy is the only dimension of jealousy that has been positively associated with high relationship quality and satisfaction can be a possible explanation for the female motivation to use benefit-provisioning strategies when experiencing elevated levels of emotional jealousy.

Despite the increasing number of research on partner retention strategies and jealousy over the last few decades, female jealousy and mate retention have received much less attention in research in comparison with male jealousy and mate retention. Also, there has been very little research that examines the relationship between these strategies and romantic jealousy, especially during the COVID-19 pandemic. As Salkicevic et al. [35] argued that mate retention behavior is stable over time and Richter et al. [36] pointed out the dispositional nature of jealousy, it is worth examining whether pandemic conditions, such as social isolation, had any effect on the relationship pattern between jealousy and mate retention.

Therefore, the main aim of the current study was to investigate the relationship between three dimensions of jealousy and mate retention strategies among women in

a romantic relationship within the age range of 19–40. We hypothesized that all three types of jealousy (cognitive, emotional, and behavioral) are positively correlated to all mate retention strategies (direct guarding, intersexual negative inducements, positive inducements, public signals of possession, and intrasexual negative inducements) [9,34]. Furthermore, we expect all three types of jealousy and relationship length to be positive predictors of both cost-inflicting and benefit-provisioning mate retention strategies, whereas age to be a negative one [7,8,10,27,30,34].

## 2. Materials and Methods

### 2.1. Participants

A total of 901 participants accessed the online survey; 108 participants were excluded for the following reasons: not having a father figure while growing up ($n = 48$), being single ($n = 45$), being a male ($n = 13$), and not being in a heterosexual relationship ($n = 2$). Additionally, cases identified as multivariate outliers ($n = 21$) were excluded from further analysis, which is discussed in the results. Hence, the final sample consisted of 772 Croatian women that were between the ages of 19 and 40 in a heterosexual romantic relationship.

Participants provided information on their gender (male, female, or other), age (year categories: 19–21, 22–25, 26–30, 31–35, and 36–40), completed education (elementary school, secondary or vocational school, undergraduate degree, graduate degree, postgraduate, or doctoral degree), relationship status (single, in a dating relationship, or married), relationship length (less than 6 months, 6 months–1 year, 1–2 years, 2–5 years, 5–10 years, or more than 10 years), living with a partner (yes or no), whether they were in a heterosexual relationship or homosexual relationship, and did they know their father or any person who represented a father figure in their life (yes or no). Sexual orientation and relationship with the father were controlled as research has shown implications to the variables of interest, as discussed in the Introduction [9,14,25,37,38]. Demographic characteristics of the sample can be seen in Table 1.

**Table 1.** Frequency distribution of demographic variables.

| Demographic Characteristics | | f (%) |
|---|---|---|
| Relationship status | Married | 185 (24.0) |
| | In non-marital relationship | 587 (76.0) |
| Age | 19–21 years old | 133 (17.2) |
| | 22–25 years old | 336 (43.5) |
| | 26–30 years old | 157 (20.3) |
| | 31–35 years old | 85 (11.0) |
| | 36–40 years old | 61 (7.9) |
| Education | Elementary school | 3 (0.4) |
| | Secondary or vocational school | 267 (34.6) |
| | Undergraduate degree | 264 (34.2) |
| | Graduate degree | 214 (27.7) |
| | Postgraduate or doctoral degree | 24 (3.1) |
| Relationship length | Less than 6 months | 62 (8.0) |
| | 6 months–1 year | 80 (10.4) |
| | 1–2 years | 117 (15.2) |
| | 2–5 years | 267 (34.6) |
| | 5–10 years | 161 (20.9) |
| | more than 10 years | 85 (11.0) |
| Cohabitation | Yes | 420 (54.4) |
| | No | 352 (45.6) |

### 2.2. Instruments

#### 2.2.1. Multidimensional Jealousy Scale

To investigate jealousy in romantic relationships, the Multidimensional Jealousy Scale [3,39] was used. This scale measures cognitive, emotional, and behavioral jealousy and consists of a total of 24 items, or 8 items in each subscale. Each item in the questionnaire refers to the person "X", who represents the participant's romantic partner. A 7-point response scale is used for each statement, with 1 indicating "Never" and 7 indicating "Always" for the cognitive and behavioral jealousy subscales, while 1 indicates "Not at all" and 7 indicates "Extremely" for the emotional jealousy subscale. The results for each subscale are formed as a linear combination, with results ranging from 8 to 56, with higher scores indicating higher levels of jealousy. Pfeiffer and Wong [3] found internal consistency for the cognitive jealousy subscale to be 0.92, 0.85 for the emotional jealousy subscale, and 0.89 for the behavioral subscale.

#### 2.2.2. Mate Retention Inventory

To investigate the use of mate retention strategies, the Mate Retention Inventory [7,40] was used, specifically the Women's Version-Heterosexual Relationship. The questionnaire consists of 104 items that describe various behaviors to retain a partner, which are grouped into 5 specific categories: direct guarding, intersexual negative inducements, positive inducements, public signals of possession, and intrasexual negative inducements. The categories can also be grouped into cost-inflicting (direct guarding, intersexual negative inducements, and intrasexual negative inducements) and benefit-provisioning (positive inducements, and public signals of possession) mate retention strategies. A 4-point scale is used to assess the frequency of performing each mate retention act, with 0 meaning never and 3 meaning often. A higher score represents more frequent use of partner retention strategies.

For the Croatian sample, the reliability coefficients for the categories of partner retention for women were as follows: direct guarding $\alpha = 0.76$, intersexual negative inducements $\alpha = 0.74$, positive inducements $\alpha = 0.76$, public displays of possession $\alpha = 0.65$, and intrasexual negative inducements $\alpha = 0.42$ [40]. Additionally, in previous research for cost-inflicting mate retention strategies, the obtained reliability was 0.84, whereas for benefit-provisioning mate retention strategies, it was 0.77 [10].

### 2.3. Procedure

An online survey was created using the Google Forms platform. It was disseminated through e-mail lists and was shared in various groups on the social network Facebook, using a snowball sampling technique. Data were collected between 15 April 2020 and 10 September 2020.

At the beginning of the questionnaire, all participants were informed of the purpose of the research and guaranteed anonymity. They were also informed that they could withdraw from participation at any time without any consequences and that their results would be used solely for the purpose of this research. Furthermore, participants were told that they could direct any queries about the research and its results to the researcher's email address. After reading and agreeing to the informed consent about the terms of participation, participants could proceed to answer the questionnaire. If they passed all elimination questions (gender, having a father figure in their lives, and heterosexual relationship), participants completed the survey and were given clear instructions before each instrument. The completion took about 15 min, and at the end of the questionnaire, participants were thanked for their responses. Participants did not receive any compensation for participation.

### 2.4. Data Analysis

The data were analyzed using SPSS (25.0). Initial data examination included the analysis of outliers, assessing the normality of distributions, and multicollinearity diagnostics. Most of the distributions did not meet normality criteria, which can be seen in

Table 2. Parametric statistical procedures were nevertheless used when possible (Pearson correlation, multiple regression analysis) because more detailed data inspection showed that the criteria suggested by Kline [41] were met.

**Table 2.** Descriptive statistics for all measured variables (N = 772).

| Variable | M | C | SD | Min. | Max. | γ1 | κ | α |
|---|---|---|---|---|---|---|---|---|
| Direct guarding | 25.51 | 24.00 | 6.15 | 18 | 55 | 16.93 | 14.03 | 0.86 |
| Intersexual negative inducements | 39.58 | 37.00 | 9.04 | 29 | 77 | 15.45 | 8.69 | 0.88 |
| Positive inducements | 54.51 | 54.00 | 9.65 | 27 | 89 | 4.20 | 1.82 | 0.84 |
| Public signals of possession | 31.15 | 31.00 | 6.62 | 15 | 51 | .79 | −1.88 | 0.81 |
| Intrasexual negative inducements | 17.29 | 16.00 | 2.20 | 16 | 34 | 35.57 | 76.82 | 0.75 |
| Cost-inflicting strategies | 82.38 | 77.50 | 15.75 | 63 | 155 | 17.05 | 12.10 | 0.92 |
| Benefit-provisioning strategies | 85.66 | 84.00 | 14.42 | 45 | 137 | 3.18 | 0.57 | 0.89 |
| Cognitive jealousy | 14.11 | 12.00 | 7.34 | 8 | 45 | 18.52 | 13.69 | 0.89 |
| Emotional jealousy | 33.55 | 35.00 | 12.74 | 8 | 56 | −2.95 | −4.66 | 0.92 |
| Behavioral jealousy | 14.27 | 12.00 | 6.46 | 8 | 42 | 20.00 | 17.90 | 0.86 |

## 3. Results

### 3.1. Data Examination and Descriptive Statistics

Analysis of outliers consisted of employing the Mahalanobis $D^2$ measure using the linear regression method to detect multivariate outliers. All cases (*n* = 21) that were detected as multivariate outliers were excluded from further analysis. After the examination of outliers, the normality of distribution was assessed using skewness (γ1) and kurtosis (κ) indices, since tests of normality tend to be significant in large samples, even when deviation from normality is minor. As suggested by Kline [41], for assessing the normality of distribution in large samples (i.e., N > 300), absolute values of skewness and kurtosis should be used. A conservative rule of thumb states that skewness should not be greater than 3 and kurtosis should not be greater than 10 to proceed with parametric statistical procedures, since the distributions are not severely non-normal. As shown in Table 2, only public signs of possession, benefit-provisioning strategies, and emotional jealousy met the parametric criteria suggested by Kline [41], although most of the variables were not distributed normally. More specifically, the non-normal distributions were positively asymmetric, except for that of emotional jealousy, indicating that participants' scores leaned toward values below the scale average. Reliability analysis showed that all measured variables had adequate psychometric properties. Descriptive statistics for all measured variables can be seen in Table 2.

### 3.2. Relationship between Types of Jealousy and Mate Retention Strategies

Pearson and Spearman correlation analyses were run to investigate the relationship between types of jealousy and mate retention strategies, depending on the deviation from normality of distributions noted in Table 3. The results showed that all three types of jealousy were positively correlated with all mate retention strategies. Regarding specific mate retention strategies, behavioral jealousy is strongly correlated with direct guarding and intersexual negative inducements, while correlations with positive inducements, public signals of possession, and intrasexual negative inducements were low. Correlations between cognitive and emotional jealousy and specific mate retention strategies were low to moderate.

**Table 3.** Correlation between types of jealousy and mate retention strategies (N = 772).

| Variable | Cognitive Jealousy | Emotional Jealousy † | Behavioral Jealousy |
|---|---|---|---|
| Direct guarding | 0.43 ** | 0.35 ** | 0.69 ** |
| Intersexual negative inducements | 0.51 ** | 0.40 ** | 0.62 ** |
| Positive inducements | 0.33 ** | 0.26 ** | 0.36 ** |
| Public signals of possession† | 0.12 ** | 0.18 ** | 0.28 ** |
| Intrasexual negative inducements | 0.33 ** | 0.21 ** | 0.35 ** |

Note. ** $p < 0.01$; † distributions are not severely not normal.

Before conducting multiple regression analysis, a Pearson and Spearman correlation matrix was calculated between age, relationship length, cognitive, emotional, and behavioral jealousy, cost-inflicting, and benefit-provisioning strategies.

As shown in Table 4, cognitive and emotional jealousy were moderately positively correlated while behavioral jealousy was strongly positively correlated with the domain of cost-inflicting strategies. The correlations between all three aspects of jealousy and the benefit-provisioning strategies were positive, but fairly low. The correlation between all three types of jealousy was moderate, as was that between cost-inflicting and benefit-provisioning mate retention strategies. Regarding age and relationship length, age was negatively correlated with all types of jealousy and mate retention strategies, whereas relationship length was negatively correlated with cognitive jealousy. All obtained coefficients regarding age and relationship length were very low and doubtfully meaningful.

**Table 4.** Correlation matrix for predictor and criteria variables (N = 772).

| Variable | 1. | 2. | 3. | 4. | 5. | 6. | 7. |
|---|---|---|---|---|---|---|---|
| 1. Age | 1 | | | | | | |
| 2. Relationship duration | 0.48 ** | 1 | | | | | |
| 3. Cognitive jealousy | −0.11 ** | −0.13 ** | 1 | | | | |
| 4. Emotional jealousy † | −0.12 ** | −0.02 | 0.38 ** | 1 | | | |
| 5. Behavioral jealousy | −0.08 * | 0.02 | 0.50 ** | 0.39 ** | 1 | | |
| 6. Cost-inflicting strategies | −0.07 * | 0.06 | 0.55 ** | 0.41 ** | 0.75 ** | 1 | |
| 7. Benefit-provisioning strategies † | −0.12 ** | 0.04 | 0.28 ** | 0.26 ** | 0.37 ** | 0.53 ** | 1 |

Note. * $p < 0.05$; ** $p < 0.01$; † distributions are not severely not normal.

As suggested by Kline [41], before conducting regression analysis, we checked if crucial criteria were met for our criterion variables, that is, cost-inflicting and benefit-provisioning mate retention strategies. Firstly, there were no multivariate outliers as mentioned before. For both criteria, the residuals were normally distributed and there was no collinearity (VIFs ranged between 1.26 and 1.45).

The results from the multiple regression analysis predicting cost-inflicting strategies are outlined in Table 5. The model is statistically significant (F(5,766) = 245.55, $p < 0.001$), explaining 61% of the criterion variance ($R^2_{adjusted} = 0.61$).

**Table 5.** Multiple linear regression predicting cost-inflicting mate retention strategies (N = 772).

| Predictor | β | t | p |
|---|---|---|---|
| Age | −0.03 | −1.07 | 0.283 |
| Relationship length | 0.07 | 2.92 | 0.004 |
| Cognitive jealousy | 0.21 | 7.95 | 0.000 |
| Emotional jealousy | 0.10 | 3.66 | 0.000 |
| Behavioral jealousy | 0.61 | 22.51 | 0.000 |

As expected, all three types of jealousy and relationship length were significant and positive predictors of cost-inflicting mate retention strategies, and behavioral jealousy was the strongest one. Age was not a significant predictor of cost-inflicting strategies.

The results from multiple regression analysis predicting benefit-provisioning strategies are outlined in Table 6. The model was statistically significant ($F_{(5,766)} = 32.82$, $p < 0.001$), explaining 15.6% of the criterion variance ($R^2_{adjusted} = 0.156$).

**Table 6.** Multiple linear regression predicting benefit-provisioning mate retention strategies (N = 772).

| Predictor | β | t | p |
|---|---|---|---|
| Age | −0.13 | −3.55 | 0.000 |
| Relationship length | 0.12 | 3.25 | 0.001 |
| Cognitive jealousy | 0.11 | 2.78 | 0.006 |
| Emotional jealousy | 0.10 | 2.86 | 0.004 |
| Behavioral jealousy | 0.26 | 6.63 | 0.000 |

As expected, all three types of jealousy and relationship length were significant and positive predictors of benefit-provisioning mate retention strategies, with behavioral jealousy being the strongest one, whereas age was a negative one.

## 4. Discussion

This study aimed to investigate the relationship between jealousy and mate retention strategies among women in a romantic relationship during the COVID-19 pandemic. As expected, it was found that all partner retention strategies were positively associated with all aspects of jealousy. Regarding specific mate retention strategies, behavioral jealousy was strongly associated with direct guarding and intersexual negative inducements, while correlations with positive inducements, public signals of possession, and intrasexual negative inducements were low. The correlations between cognitive and emotional jealousy and specific mate retention strategies were low to moderate. The findings of this research indicate that a stronger experience of jealousy, especially behavioral jealousy, can be expected to result in more frequent use of all partner retention strategies. This is not surprising because, according to evolutionary theory, the emotion of jealousy serves the same function as partner retention strategies [1]. The perceived likelihood of a partner's infidelity has previously been associated with partner retention strategies, with studies showing that these strategies are more commonly employed when participants believe that there is a likelihood of their partner being unfaithful [42]. The perceived likelihood of a partner's infidelity itself can be explained as a result of the evaluation process an individual undergoes when appraising a partner's behavior [14], which is consistent with the dynamic functional model of jealousy [34] within the social–cognitive theory of jealousy [11,12,14]. The model states that, once a threat, either to the relationship or the self, is perceived, an individual is motivated to act upon it in order to preserve the relationship. The cognitive and emotional dimensions of jealousy are directly related to the suspicion of a partner's infidelity and the emotions that arise from it, so an association between them and the frequency of using partner retention strategies is expected when an individual values preserving a romantic relationship with their partner. Direct guarding and behavioral jealousy both serve to prevent a partner from leaving the romantic relationship or cheating on the partner, and Buss [26] refers to such behaviors as mate guarding. The association of jealousy with intersexual negative inducements that range from intentionally provoking jealousy in a partner, humiliating the partner in front of a rival, emotional and sexual coercion, to insulting the partner and physical violence, is consistent with research on partner violence. Arnocky et al. [43] found that such tactics are used to punish a partner and discourage the partner from committing infidelity, and individuals with strong feelings of jealousy are prone to reacting in this way [1].

Only one additional study, to our knowledge, has been conducted that examined the relationship between these constructs, and its findings also indicate that all aspects of partner retention are positively correlated with all three dimensions of jealousy [10]. Conar [10] obtained a very similar correlation pattern regarding the relationship between jealousy dimensions and mate retention strategies, including low correlations between

jealousy dimensions and the category of public displays of possession. This indicates that this finding cannot be explained as the effect of social isolation, As Conar's study [10] was conducted prior to the COVID-19 pandemic.

Regarding cost-inflicting and benefit-provisioning mate retention strategies, multiple regression analysis showed that both could be predicted by jealousy and relationship length, whereas age was a negative predictor of benefit-provisioning strategies only. Cognitive and emotional jealousy were moderately related, while behavioral jealousy was strongly related to the domain of cost-inflicting strategies. On the other hand, the relationship between all three aspects of jealousy and the benefit-provisioning domain was fairly weak. Additionally, jealousy, age, and relationship length explained a substantial proportion of cost-inflicting variance, whereas their effects on benefit-provisioning strategies were small in size. Therefore, it can be concluded that all three dimensions of jealousy are more important for engaging in interpersonally damaging, risky, and aggressive acts (e.g., monopolizing a romantic partner's time, emotionally manipulating one's mate, and threatening to hit an intrasexual rival) than for positive inducements (e.g., gift giving, appearance enhancement, performing sexual favors, and being more attentive and affectionate) and public signals of possession (e.g., holding hands and talking favorably about one's romantic partner to others) [7]. This is in line with the findings that jealousy is directly and firmly associated with aggressive behavior in women, especially in the context of mate retention [44]. Also, it seems that jealousy can be considered a positive and adaptive emotion because it predicts attentive and affectionate romantic behavior, to some extent [2,37], which is associated with favorable relationship outcomes [7,28]. This is also in line with the dynamic functional model of jealousy [34], since the model proposes that the final outcomes of elicited jealousy can be both beneficial and detrimental to the maintenance of the relationship.

Regarding age and relationship length, the results showed that, even if there is a significant association, the effects are very low and doubtfully meaningful. Age was a negative predictor of benefit-provisioning mate retention strategies only, indicating that as women age, they are less prone to using these strategies. Regarding relationship length, it was a positive predictor for both cost-inflicting and benefit-provisioning mate retention strategies, indicating that women who are in longer relationships are more motivated to use both strategies. Since relationship length was not associated with cost-inflicting nor benefit-provisioning strategies, this was clearly a suppression effect via jealousy. Hence, it is worth considering the general patterns of jealousy adopted by women, which become more firmly established as they age. The reason for this may be the higher number of romantic relationships among older women. In other words, older women are more likely to have been in more intimate relationships and have therefore experienced more abandonments than younger women. It is reasonable to assume, therefore, that they will exhibit more signs of jealousy in relationships, especially if more resources have been invested and especially if they share children with their partner. Regardless, as people grow older, entering a romantic relationship is perceived as a more serious matter and partners are more motivated to retain each other. It is also important to note that youth is an important signal of fertility and relative health, and is also the most desirable trait in women from an evolutionary point of view [8,42]. Additionally, evolutionary theories suggest that the decline in reproductive capacity for women with age influences jealous behaviors because a woman's value as a partner decreases with age and the person feels more threatened [45]. From the social–cognitive point of view, Chung and Harris [34] agree that individuals in more committed long-term relationships experience higher levels of jealousy, but the explanation is somewhat different. They argue that, as the relationship lasts longer, people perceive a higher value of the relationship and consequently greater loss if terminated. Interestingly, intensified jealousy is directed only to positive, constructive, and beneficial outcomes and behavior in order to maintain relationship commitment and satisfaction, as well as the relationship itself.

Lantagne and Furman [46] conducted an interesting longitudinal study on the development of romantic relationship characteristics with age and relationship length. The

findings showed that long-term adolescent relationships were characterized by mutual support, elevated levels of support, negative interactions, control, and jealousy. Over time, long-term relationships continued to have a high level of support, but the level of negative interactions, control, and jealousy decreased. Guerrero and Afifi [4] found that individuals motivated to maintain their current relationship were prone to expressing behavioral jealousy. Older women in longer and more committed relationships may be equally or even more motivated to maintain their intimate relationships than younger women who have just begun romantic relationships [7]. The reason why this study showed no statistically significant age effect on cost-inflicting mate retention strategies might lay in the fact that, during the COVID-19 pandemic, both partners had fewer opportunities to socialize in the sense that social distancing and isolation may have resulted in a reduced perception of relationship threat. Another possible explanation is the diminished statistical power due to the uneven number of participants in measured age groups. The results also showed that, as age increased, there was a slight decline in the usage of mate retention strategies, which is consistent with De Miguel and Buss' [27] findings that younger women in less committed relationships generally used all partner retention strategies more frequently. The explanation for this might be that younger people in newer relationships feel more insecure and have less trust in their romantic partner and therefore use partner retention tactics more often. This is consistent with the dynamic functional model of jealousy because Chung, and Harris suggest that, at early relationship stages, commitment and satisfaction serve as a protective mechanism against doubt and suspicion that something or someone is a threat. Hence, jealousy emerges and directs one's behavior in less committed and shorter relationships [34].

## 5. Conclusions

The results showed that female jealousy and mate retention are strongly associated, especially when considering interpersonally damaging and aggressive behavior to retain a partner, and that jealousy can be considered positive and adaptive when associated with affectionate and attentive behavior towards a partner.

## 6. Limitations and Future Research

This study has several limitations. First, the sample is not representative as the study was conducted online and thus the generalizability of the findings is questionable. There is also an issue with the uneven distribution of the sample by age, education level, and duration of the relationship, which is very likely to have influenced the distribution of certain variables and the relationship between them. Furthermore, the observed relationships among variables in terms of correlation did not imply causality, and as this is not a longitudinal study, any conclusion on the direct effects of aging or COVID-19 restrictions on the relationship between jealousy and mate retention strategies should be avoided, especially when taking into account that we tested general relationship patterns between jealousy and mate retention, not the absolute values of the exhibited behavior or experience. Considering age and relationship length, the sample size should also be considered, since it can be a reason why some of the obtained coefficients reached statistical significance, but also the way both were measured. We had groups different in size, which very likely decreased statistical power. Furthermore, these findings and explanations should be considered critically, given that they partly rely on evolutionary theory, which practically suggests that only young and beautiful women are desirable partners. Future research should include both dispositional (e.g., neuroticism, self-esteem, and dark personality traits) and situational (e.g., infidelity experience, relationship satisfaction, perceived relationship quality, cultural norms, and context) variables to gain a deeper and more comprehensive insight into partner retention behavior [7,10,28,35,36].

**Author Contributions:** Conceptualization, P.D. and D.V.; methodology, P.D. and D.V.; formal analysis, P.D. and L.A.; data curation, P.D. and L.A.; writing—original draft preparation, P.D.; writing—review and editing, L.A.; supervision, D.V.; All authors have read and agreed to the published version of the manuscript.

**Funding:** This research received no external funding.

**Institutional Review Board Statement:** This study was conducted according to the guidelines of the Declaration of Helsinki and approved by the Institutional Review Board (or Ethics Committee) of the Department of Psychology, Faculty of Croatian Studies, University of Zagreb (protocol code 380-58/506-20-0001 and date of approval 14 April 2020).

**Informed Consent Statement:** Informed consent was obtained from all subjects involved in this study.

**Data Availability Statement:** Data will be made available upon reasonable request to the corresponding author.

**Conflicts of Interest:** The authors declare no conflict of interest.

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
