# Peer review of "The Relationship between Jealousy and Mate Retention Strategies in Romantic Relationships among Women during the COVID-19 Pandemic"

_ejihpe, doi:10.3390/ejihpe13120199_

Round 1

Reviewer 1 Report

Comments and Suggestions for Authors

This is a well-written and thoughtful manuscript that deals with an important issue in romantic relationships: the role of jealousy in mate retention.

The authors are aware of the vast psychological literature, and their empirical findings are quite relevant to previous studies, while having some novel findings.

I would have liked to have seen some discussion in the article of the main issue underlying jealousy and mate retention, namely, exclusivity. In this respect, it would be interesting to explore the relationship between jealousy mate retention in polyamory and other consensual nonmonogamous relationships.

Author Response

Thank you very much for taking the time to review this manuscript and for your comments. We included your remark on non-monogamous relationships within the jealousy-evoking mechanism in the introduction.

Reviewer 2 Report

Comments and Suggestions for Authors

The topic of the article is interestingo. However, the article lacks a sociocultural perspective, both in the introduction and in the interpretation of the results. It focuses on evolutionary approaches, ignpraying the importance of gender socialization. A further review of the literature would be needed.

In the introduction section, the authors allude to the fact that spending more time at home with one's partner during confinement could increase jealousy, however the time covered in the research includes the time of confinement, but also the post-confinement, so the results may be mixing different realities.

The sample size is too small to be able to draw generalizations from the population, and differences in group size diminish statistical power.

The use of sex and the fact of having a father figure during their growth as exclusion criteria are not sufficiently substantiated.

Conclusion section should be before the limitations and future research section.

Author Response

Thank you for taking the time to review this manuscript and for all your comments. We attached our responses in pdf file.

Reviewer 3 Report

Comments and Suggestions for Authors

The article entitled The Relationship between Jealousy and Mate Retention Strategy- 2 gies in Romantic Relationships among Women during the 3 COVID-19 Pandemic, complies with the journal's editorial requirements. Being a topical issue, this approach was also necessary. Therefore, the instruments used in the research to operationalize the research objectives are identified. Theoretical foundation and bibliographic resources are accepted as study as well as research method. Congratulations to the authors.

Author Response

Thank you for taking the time to read this manuscript and for your comments.

Reviewer 4 Report

Comments and Suggestions for Authors An article about the simple, eternal and constant, about intrapersonal conflict, about lack of confidence in yourself and your partner. It's simple. No further comments.

Author Response

Thank you for taking the time to read this manuscript and for your comments. We included an additional framework for explaining the jealousy-evoking mechanism in both the introduction and discussion.

Round 2

Reviewer 2 Report

Comments and Suggestions for Authors

All the comments have been adressed.